# Hyperglycemia compromises Rat Cortical Bone by Increasing Osteocyte Lacunar Density and Decreasing Vascular Canal Volume

Birol Ay [1], Kushagra Parolia[2], Robert S. Liddell[3], Yusheng Qiu[4], Giovanni Grasselli[4], David M.L. Cooper[2] & John E. Davies[1,3]*

Uncontrolled diabetes is associated with increased risk of bony fractures. However, the mechanisms have yet to be understood. Using high-resolution synchrotron micro-CT, we calculated the changes in the microstructure of femoral cortices of streptozotocin-induced hyperglycemic (STZ) Wistar Albino rats and tested the mechanical properties of the mineralized matrix by nanoindentation. Total lacunar volume of femoral cortices increased in STZ group due to a 9% increase in lacunar density. However, total vascular canal volume decreased in STZ group due to a remarkable decrease in vascular canal diameter ($7 \pm 0.3$ vs. $8.5 \pm 0.4 \, \mu m$). Osteocytic territorial matrix volume was less in the STZ group ($14,908 \pm 689 \, \mu m^3$) compared with healthy controls ($16,367 \pm 391 \, \mu m^3$). In conclusion, hyperglycemia increased cellularity and lacunar density, decreased osteocyte territorial matrix, and reduced vascular girth, in addition to decreasing matrix mechanical properties in the STZ group when compared with euglycemic controls.

[1] Institute of Biomaterials and Biomedical Engineering, University of Toronto, Toronto, ON, Canada. [2] Department of Anatomy, Physiology and Pharmacology, University of Saskatchewan, Saskatoon, SK, Canada. [3] Faculty of Dentistry, University of Toronto, Toronto, ON, Canada. [4] Department of Civil Engineering, University of Toronto, Toronto, ON, Canada. *email: jed.davies@utoronto.ca

Diabetes mellitus (DM), a chronic disease characterized by high blood glucose (hyperglycemia), is an escalating global issue. According to the International Diabetes Federation's report, there are currently an estimated 425 million diabetics worldwide, of which 212 million have not been diagnosed[1]. Importantly, uncontrolled diabetes has been associated with an increased risk of dental implant failure[2] and vertebral/hip fractures[3].

It is generally agreed that bone quality is compromised in hyperglycemic compared with euglycemic bone[4–7], but the reasons are poorly understood. Reports of decreased implant stability[8] and retention[9,10] in hyperglycemic subjects support the notion of compromised bone quality. It has been demonstrated that bone healing delays[11], growth plate thickness reduces[12], cortical porosity increases due to bone loss[13,14], and the cross-linking patterns of bone collagen changes with advanced glycation end products (AGEs)[15–17] in hyperglycemic subjects. Yet, little has been done to elucidate the changes in bone cell density and vascular architecture in hyperglycemic bone.

de Mello-Sampayo et al.[18] reported an increase in the number and size of bone lacunae in the lumbar vertebrae (L4) of 5-month streptozotocin (STZ)-induced hyperglycemic rats from scanning electron microscopy (SEM) images but without quantitative confirmation. Indeed, manual counting of lacunar number in their images contradicts their conclusion. Villarino et al.[19] reported a decrease in lacunar density, although they provided no histological images, and their lacunar volume measurements were based on area measurements. On the contrary, Kerckhofs et al.[20] used microfocus X-ray computed tomography (micro-CT) to investigate three-dimensional (3D) bone microstructure and reported an insignificant difference in lacunar density between the tibial mid-diaphysis of high-fat diet (HFD)-fed male C57BL/6 mice (14-week hyperglycemic) and their age-matched healthy controls. However, there was a decrease in vascular canal volume in HFD mouse bone compared with control[20]. Nevertheless, as the authors report, the HFD mouse model does not allow the effects of hyperglycemia, insulin resistance, excess body weight, and aging on bone tissue to be separately distinguished. In a similar case, Karunaratne et al.[21]

investigated Crh$^{-120/+}$ female mouse tibial mid-diaphyses with synchrotron radiation (SR) micro-CT and reported a decrease in vascular canal and osteocyte lacunar density in Crh$^{-120/+}$ mouse bone compared with control. As Crh$^{-120/+}$ mice are obese, hypercorticosteronaemic, and hyperglycemic, it is difficult to attribute the changes observed in bony structure to either hyperglycemia or osteoporosis independently. Even though there is no single animal model that can show all the features of diabetic skeletal fragility in humans as previously discussed by Fajardo et al.[22], the effects of metabolic changes on bone microstructure need to be investigated in a simpler animal model.

Our group has previously used an STZ-induced hyperglycemic rat model, which is neither osteoporotic nor obese, and reported an increase in osteocyte lacunar density along with decreased mineralization in newly formed peri-implant bone[23]. Based on these previous qualitative findings, we hypothesized that osteocytic territorial matrix volume decreases due to a reduction in total mineralized matrix volume, which would result in increased osteocyte lacunar density in STZ-induced hyperglycemic rats. Based on our previous studies of implant osseointegration, we chose two time points: 1 month, by which time bone healing is complete[23], and 3 months, by which time significant bone remodeling should have occurred. The main purpose of the present study was to address this hypothesis by quantifying osteocyte lacuna number and bone matrix volume using high-resolution SR micro-CT to demonstrate the mechanism by which osteocyte lacunar density could increase in hyperglycemic rats. We also calculated the changes in vascular canal volume to investigate the effects of metabolic changes on bone vasculature. We found that hyperglycemia increases lacunar density due to a reduction in osteocytic territorial matrix volume but decreases total vascular canal volume due to a decrease in canal diameter.

## Results

**Less femoral length and cortical size in STZ group.** Rat femora appeared smaller in the STZ group than their healthy counterparts (Fig. 1a). Indeed, femoral length was significantly less in the

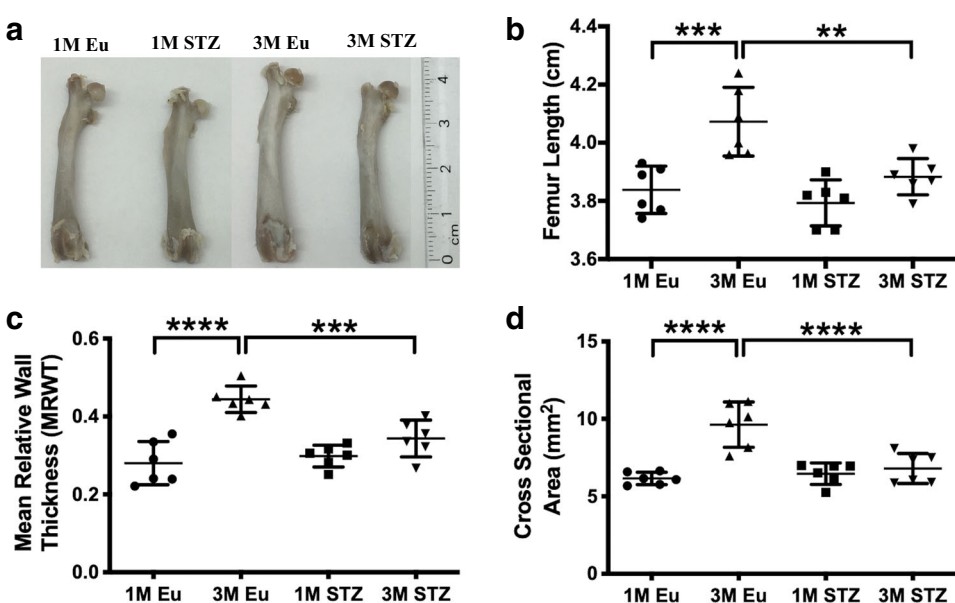

**Fig. 1 Physical features of rat femora from 1-month (1M) and 3-month (3M) euglycemic (Eu) and STZ-induced hyperglycemic (STZ) groups.**
**a** Macroscopic images, **b** femur length, **c** mean relative wall thickness (MRWT), and **d** cross-sectional area of cortical rat femora. Although there was no significant difference in between euglycemic and STZ groups in the measured parameters at 1 month, femur length, cross-sectional area, and MRWT of cortical rat femora were remarkably lower in STZ group compared with euglycemic controls (two-way ANOVA and post-hoc Newman–Keuls, mean ± SD, *$p < 0.05$, **$p < 0.01$, ***$p < 0.001$, and ****$p < 0.0001$, $n = 6$ animals).

STZ group (3.88 ± 0.06 cm) than the controls (4.10 ± 0.11 cm) at 3 months ($p < 0.01$), although there was no statistical difference at 1 month (3.79 ± 0.07 and 3.83 ± 0.08 cm, respectively) (Fig. 1b). Both mean relative wall thickness (MRWT) (0.34 ± 0.04 vs. 0.44 ± 0.03, $p < 0.001$) and the cross-sectional area of cortical bone (6.79 ± 0.90 vs. 9.63 ± 1.40 mm², $p < 0.0001$) were less in the STZ group at 3 months, although such differences were also not evident at 1 month (Fig. 1c, d). Importantly, although the femoral length, MRWT, and the cross-sectional area of femoral cortical bone significantly increased ($p < 0.01$, $p < 0.0001$, and $p < 0.0001$, respectively), with time, in euglycemic rats, there was only a small increase in the same parameters in the STZ group (Fig. 1b–d).

**Greater total lacunar volume due to higher lacunar density.** Total lacunar volume decreased from 1 to 3 months in both euglycemic and STZ groups. This was mirrored in the measurements of lacunar density. In detail, total lacunar volume was greater (+12%, $p < 0.05$) in femoral cortices of STZ compared with that in euglycemic animals at 3 months, although this difference was not evident at 1 month (Fig. 2a). Similarly, lacunar density was slightly greater at 1 month in STZ compared with the euglycemic group (76,767 ± 6525 vs. 71,767 ± 3465) and this difference became statistically significant by 3 months (67,195 ± 3206 and 61,126 ± 1457 respectively, $p < 0.05$) (Fig. 2b). Mean lacuna volume did not change among the groups at either 1 month (389 ± 27 vs. 403 ± 38 μm³) or 3 months (373 ± 6 vs. 359 ± 22 μm³) (Fig. 2c). Lacunar sphericity was also similar among the groups at 1 month (0.90 ± 0.006 vs. 0.90 ± 0.009) and 3 months (0.91 ± 0.002 vs. 0.91 ± 0.01) (Fig. 2d).

**Greater lacunar density in the cortices of STZ group.** Increased lacunar density was visualized in the SEM images of freeze fractured specimens (Fig. 3a–d) and histological sections of demineralized rat femora (Fig. 3e–h) in STZ compared with the euglycemic group. Indeed, osteocyte lacuna number/bone area was statistically significantly higher in STZ group at both 1

($p < 0.05$) and 3 months ($p < 0.01$) when the osteocyte lacunae were counted manually and normalized to the bone area in the SEM images at the same magnification (Fig. 3i). Similarly, osteocyte lacuna number/bone area in the histological sections was statistically significantly greater ($p < 0.05$) in STZ compared with the euglycemic group (Fig. 3j). Increased osteocyte lacunar density in hyperglycemic bone is illustrated in Fig. 3k.

**Less total canal volume due to smaller canal diameter.** In both 3D volume rendering and isometric projections of 3D reconstructions, vascular canals appeared to cover less area in the images of femoral cortices in STZ compared with the euglycemic group at both 1 month and 3 months. In particular, narrower vascular canal segments were notable in the STZ group (Fig. 4). Quantitative analysis of these images demonstrated that vascular canal volume was significantly lower at both 1 month (−37%, $p < 0.05$) and 3 months (−25%, $p < 0.05$) in STZ group compared with their euglycemic controls (Fig. 5a). Mean vascular segment volume was also significantly lower at both 1 month (14,760 ± 5192 vs. 24,120 ± 9307 μm³, $p < 0.05$) and 3 months (6293 ± 1332 vs. 10,020 ± 1988 μm³, $p < 0.05$) in STZ than in the euglycemic group (Fig. 5b). However, there was no significant difference in vascular density and canal length among the groups (Fig. 5c, d). On the other hand, the average diameter of vascular canal segments significantly decreased at both 1 month (10.8 ± 2 vs. 13.4 ± 2 μm, $p < 0.05$) and 3 months (7 ± 0.3 vs. 8.5 ± 0.4 μm, $p < 0.05$) in STZ group compared with healthy controls (Fig. 5e).

**Lower total and territorial matrix volume in STZ group.** Total matrix volume was similar in both STZ and euglycemic groups at 1 month and increased in each group by 3 months. Indeed, total matrix volume was statistically significantly lower (−0.4%, $p < 0.05$) in the STZ group than in healthy controls at 3 months, whereas there was no statistically significant difference among the groups at 1 month (Fig. 6a). Mineralized matrix volume per osteocyte lacuna was also significantly less ($p < 0.01$) in

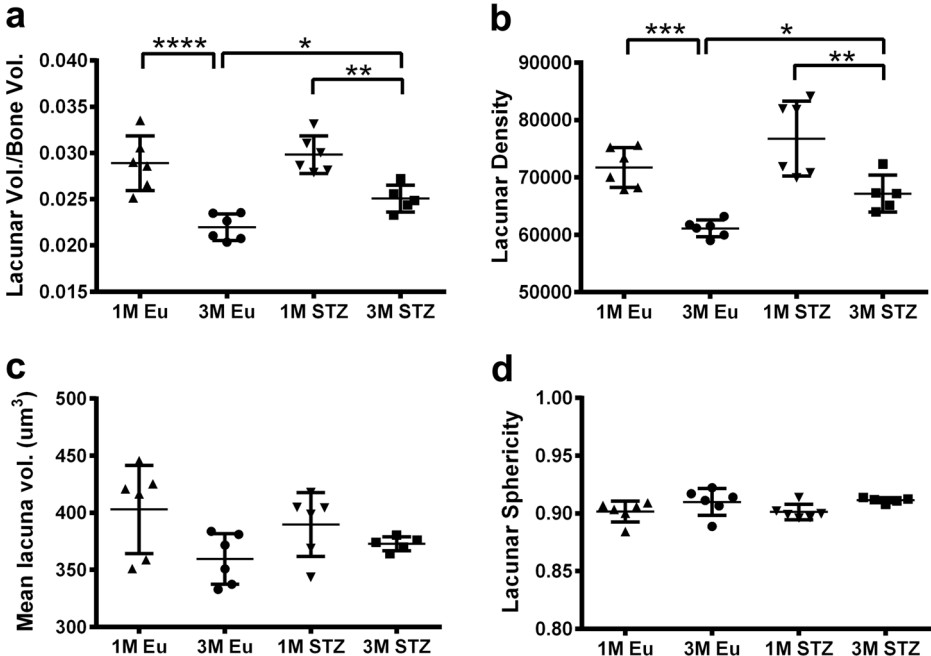

**Fig. 2 Lacunar parameters of rat cortical femora. a** Total osteocyte lacunar volume per bone volume, **b** lacunar density (lacuna number/mm³), **c** mean lacuna volume, and **d** lacunar sphericity. Total lacunar volume and lacunar density were significantly greater in STZ group at 3 month, whereas the same parameters were slightly higher in STZ group compared with euglycemic controls at 1 month. Mean lacuna volume and lacunar sphericity did not change among the groups (two-way ANOVA and post-hoc Newman–Keuls, mean ± SD, *$p < 0.05$, **$p < 0.01$, ***$p < 0.001$, and ****$p < 0.0001$, $n = 5$-6 animals).

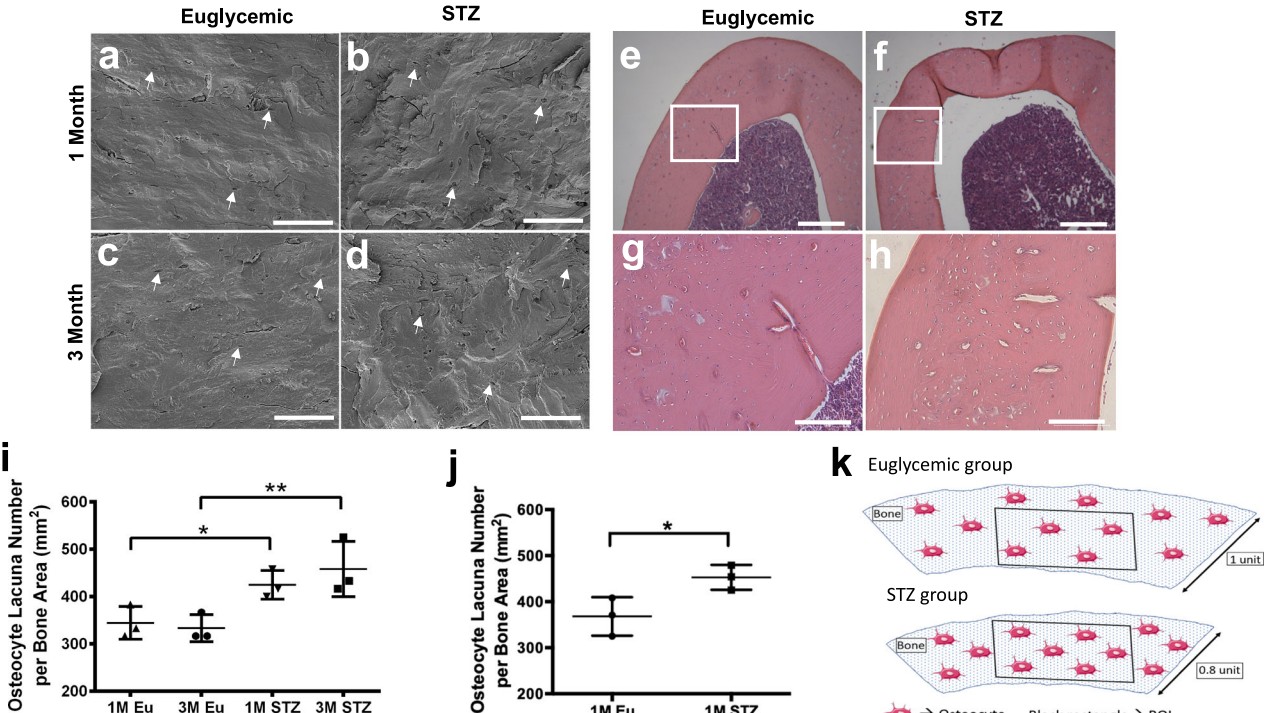

**Fig. 3 Visualization of increased lacunar density in femoral cortices of STZ group. a–d** Representative scanning electron microscopy (SEM) and **e–h** H&E-stained histology images, **i** quantification of osteocyte lacuna number in the SEM images in **a–d**, **j** quantification of osteocyte lacuna number in the histology images in **e–h**, **k** illustration of increased cellular density in the femur cortices of STZ group. In both SEM and histology images, femoral cortices of STZ group appeared more cellular than the euglycemic group (white arrows indicate osteocyte lacunae, two-way ANOVA and post-hoc Newman–Keuls for **i**, Student's *t*-test for **j**, mean ± SD, *$p < 0.05$, **$p < 0.01$, ***$p < 0.001$, and ****$p < 0.0001$, $n = 3$ animals, scale bars represent 100 μm (**a–d**, **g**, **h**) and 300 μm (**e**, **f**)).

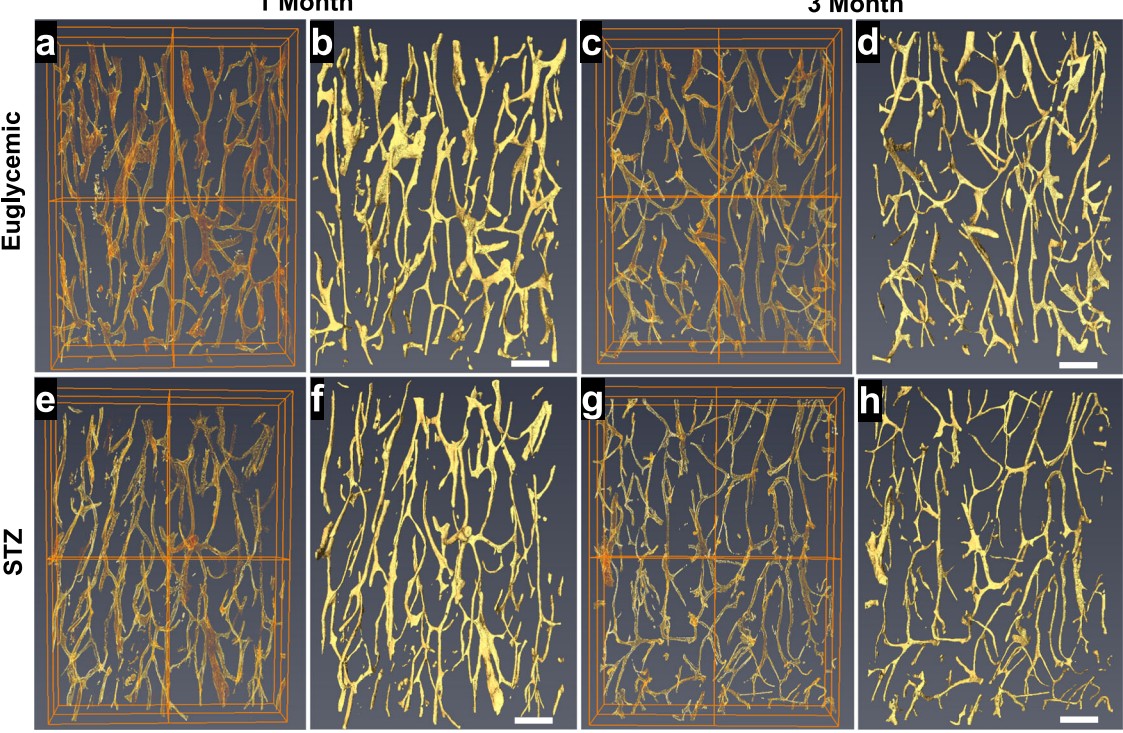

**Fig. 4 Representative 3D volume rendering and isometric projections of 3D reconstructions of vascular canals in the femoral cortices of euglycemic and STZ groups. a**, **b** 1-Month euglycemic, **c**, **d** 3-month euglycemic, **e**, **f** 1-month STZ, and **g**, **h** 3-month STZ groups. Decreased vascular canal volume in STZ group is evident in both 3D volume rendering and isometric projections of 3D reconstructions (scale bars represent 200 μm).

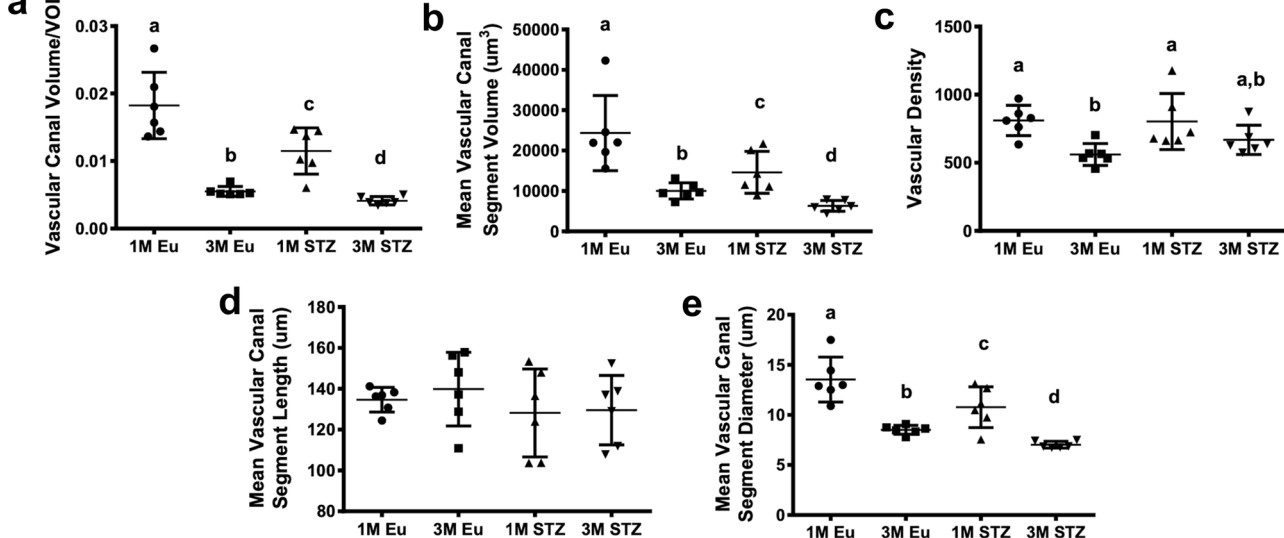

**Fig. 5 Vascular canal parameters of rat cortical femora. a** Vascular canal volume per volume of interest (VOI), **b** mean vascular canal segment volume, **c** vascular density, **d** mean vascular canal segment length, and **e** mean vascular canal segment diameter. Vascular canal volume significantly decreased due to a significant reduction in canal diameter in the femoral cortices of STZ compared with euglycemic group at both 1 month and 3 months (linear regression for **a**, **b**, **e**; two-way ANOVA and post-hoc Newman–Keuls for **c**, **d**; mean ± SD, statistical significances ($p < 0.05$) between the groups are indicated by a–d, $n = 6$ animals).

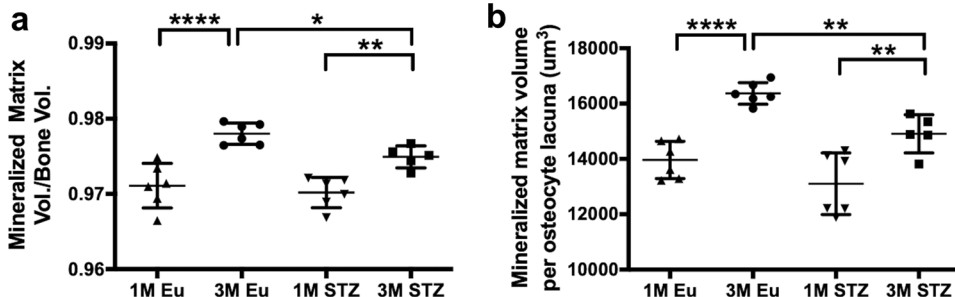

**Fig. 6 Mineralized matrix-related parameters of rat cortical femora. a** Matrix volume per bone volume, **b** matrix volume per osteocyte lacuna. Both matrix volume per bone volume and matrix volume per osteocyte lacuna were significantly decreased in STZ group compared with euglycemic group in 3-month samples, whereas there was no significant difference in both parameters at 1 month (two-way ANOVA and post-hoc Newman–Keuls, mean ± SD, *$p < 0.05$, **$p < 0.01$, ***$p < 0.001$, and ****$p < 0.0001$, $n = 5$–6 animals).

STZ ($14908 \pm 689\ \mu m^3$) than in the euglycemic group ($16,367 \pm 391\ \mu m^3$) at 3 months, but not remarkably different between STZ and euglycemic groups at 1 month ($13,105 \pm 1113$ vs. $13,962 \pm 674\ \mu m^3$) (Fig. 6b).

**Compromised cortical bone matrix quality of STZ group**. Nanoindentation tests demonstrated that the bone matrix of STZ group was less stiff and demonstrated a lower resistance to deformation compared with control at 3 months. Hardness ($-14.6\%$, $p < 0.01$) and elastic modulus ($-31\%$, $p < 0.05$) were significantly lower in the matrix of STZ than euglycemic group at 3 months. However, the matrix quality was not measurably different in 1-month samples between STZ and euglycemic groups (Fig. 7a, b).

## Discussion

Uncontrolled diabetes is known to increase the risk of bone fracture and implant failure, but the underlying reasons have yet to be fully discovered. Although it is known that the changes in bone microstructure have an effect on bone mechanics, the effects of hyperglycemia on bone microstructure have not been described in any detail. In the present study, we investigated the

changes in the lacunar density and vascular architecture in STZ-induced hyperglycemic rat femora with high-resolution SR Micro-CT to elucidate whether hyperglycemia changes the bone microstructure. Our results demonstrated that hyperglycemia increased the osteocyte lacunar density by decreasing the osteocytic territorial matrix volume, and thus increasing the cellularity of cortical bone. Moreover, total vascular canal volume decreased due to a decrease in vascular canal diameter, the total matrix volume was lower, and the matrix quality was compromised in STZ compared with the euglycemic control group.

Increased cellular density in hyperglycemic animals has been previously reported in other tissues: Spencer et al.[24] demonstrated a small increase in glomerular cellularity, due to increased proliferation of tubular epithelial cells, along with a slight increase in the extracellular matrix of the glomerular tufts in STZ-induced diabetic BALB mouse kidney. In another study, de Oliveira et al.[25] reported greater density of fibrocytes along with the increased density of type I collagen in the Achilles tendons of STZ-induced hyperglycemic rats. However, the absolute thickness of the Achilles tendons of hyperglycemic rats was lower than those of euglycemic controls[25]. Taken together, although the increased cellular density was associated with the increased cellular proliferation in the studies of both Spencer et al.[24] and de

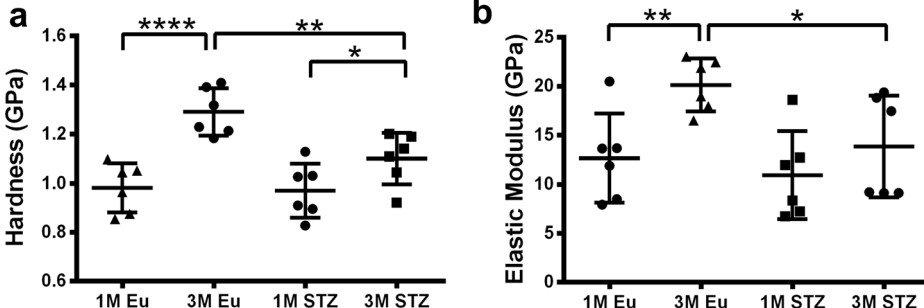

**Fig. 7 Tissue mechanical properties of mineralized matrix in STZ and euglycemic rat femoral cortices. a** Hardness and **b** elastic modulus. Bone matrix was less stiff and showed a lower resistance to deformation in STZ compared with euglycemic group in 3-month samples. However, there was not a significant change in the matrix mechanical properties of 1-month samples (two-way ANOVA and post-hoc Newman–Keuls, mean ± SD, *$p < 0.05$, **$p < 0.01$, ***$p < 0.001$, and ****$p < 0.0001$, $n = 6$ animals).

Oliveira et al.[25], in the present study, increased lacunar density was related to the decreased osteocytic territorial matrix volume.

Using high-resolution micro-CT, it has been reported that HFD mice and their euglycemic controls demonstrated similar values of total lacunar volume, lacunar density, and lacunar diameter in their mid-shaft tibiae[20]. In another study, less lacunar density in Crh$^{-120/+}$ mouse tibiae compared with controls has been demonstrated[21]. Our analyses, to the contrary, showed greater total lacunar volume in STZ-induced hyperglycemic rats due to increased lacunar density, without a change in either mean lacuna volume or lacunar sphericity (Fig. 2). There could be two potential reasons for these apparently divergent findings. First, the animal models chosen in the previous studies[20,21] showed multiple pathologies. For instance, the hyperglycemic Crh$^{-120/+}$ mice employed by Karunaratne et al.[21] were also osteoporotic and exhibited considerable interconnected porosity that was larger than the vascular canals and osteocyte lacunae we report herein. These pores clearly contributed to decreased lacunar density in the tibial cortices of Crh$^{-120/+}$ mice. Second, the anatomical locations of bones investigated, and the relative age of species, in the previous studies[20,21] were different to those in our work. Both Kerckhofs et al.[20] and Karunaratne et al.[21] made observations in the mid-shaft of the mouse tibiae, while we investigated the distal femoral rat metaphysis, which grew significantly during the hyperglycemic state and was, presumably, more severely affected by hyperglycemia than the diaphyseal tibial cortex. As for the ages of animals, although Kerckhofs et al.[20] and Karunaratne et al.[21] investigated 22- and 26-week-old mouse bone, respectively, we investigated 15- and 23-week-old rat bone. In all cases, the animals were killed at about 6 months of age, making the rats in our study relatively younger.

Even though lacunar morphology could be estimated from the morphology of osteocytes at the nano-scale using confocal laser scanning microscopy[26,27], direct quantification of lacunar morphology with SR micro-CT using the BMIT beamline at the Canadian Light Source (Saskatoon, SK, Canada) is currently limited to simple measures of orientation[28]. Therefore, a limitation of the current study is that the resolution was not sufficient to meaningfully quantify the lacunar shape: thus, possible differences in lacunar sphericity between the hyperglycemic and euglycemic groups may have been missed (Fig. 2d).

Our SR Micro-CT analyses demonstrated that both total matrix volume and osteocytic territorial matrix decreased in the STZ group (Fig. 6). This implies that the osteoblasts of the STZ group produced less matrix than those of euglycemic group. In this context, three different molecular pathways, by which hyperglycemia decreases matrix production by osteoblasts have been reported in the literature. Incubation of primary calvarial rat osteoblasts in osteogenic medium supplemented with 25.5 mM

glucose for 25 days induced reactive oxygen species-stimulated phosphatidyl inositol 3-kinase/Akt pathway and decreased the expressions of runt-related transcription factor 2, osteocalcin (OCN), and collagen I in primary osteoblasts[29]. High glucose (33 mM) impaired estrogen receptor-α transcription activity by inhibiting β-catenin signaling in MC3T3-E1 osteoblastic cells after 9-days of culture in osteogenic medium, and decreased the expressions of alkaline phosphatase (ALP), bone sialoprotein, and OCN[30]. The role of the interaction between AGEs and their cell receptor (RAGE) in decreased matrix production by osteoblasts have been investigated since AGEs increase in hyperglycemic subjects and pathologically accumulate in serum and stromal tissues:[31] Meng et al.[32] demonstrated that artificially glycated bovine serum albumin (150 μg ml$^{-1}$) increased the gene and protein expressions of RAGE, while decreasing both ALP and OCN gene expression via the RAGE/Raf/MEK/ERK signaling pathway in human fetal osteoblastic cells (hFOB1.19).

It has been demonstrated that vascular canal density in tibial cortices decreased in Crh$^{-120/+}$ mice, which are obese, hyper-corticosteronaemic, and hyperglycemic[21]. In another study, a decrease in total vascular canal volume due to decreased vascular canal density in HFD mouse bone was reported, without a remarkable change in canal diameter[20]. In our analyses, total vascular canal volume in the STZ group was also decreased but this was related to the decreased diameter of vascular canals rather than vascular canal density and length.

Decreased vascular canal volume implies that hyperglycemia compromises vascular structure. Indeed, deleterious effects of hyperglycemia on vascular structure have been shown in several studies: It is reported that high glucose increased senescence and impaired the in vitro tube forming capacity of endothelial progenitor cells (EPCs) isolated from peripheral blood of healthy individuals[33]. Furthermore, human peripheral blood derived EPCs demonstrated decreased in vitro vessel-forming capacity due to a decrease in angiopoietin 1 expression, the gene regulating endothelial cell (EC) survival and vascular maturation, in the presence of high glucose[34]. In another study, co-culture of human ECs and smooth muscle cells in a spheroid, in high glucose, resulted in decreased vessel length and diameter due to the suppressed notch1, the pathway related to the formation and maintenance of the vascular system in ECs[35]. Importantly, the reduction in vascularization could presumably affect hyperglycemic bone by three potential mechanisms: first, as angiogenesis precedes osteogenesis, a decrease in the volume of blood vessels could result in an insufficient nutrient and oxygen supply to the osteoblasts of newly forming bone and might result in decreased matrix production by osteoblasts. Second, decreased nutrient and oxygen supply in hyperglycemic bone could trigger osteocyte apoptosis[36]. Thereafter, increased osteocyte apoptosis could induce osteoclast

precursor recruitment[37] and adhesion[38] and eventually increase bone resorption in the longer term[39,40]. Third, decreased angiogenesis could also result in the recruitment of less perivascular cells, which are the precursors of osteogenic cells[41]. These predicted mechanisms are worth further investigation since, in human subjects, they could be contributing factors to the increased cortical porosity due to bone loss reported in hyperglycemics[42,43].

Our present findings on compromised matrix quality of femoral cortices in the STZ group are supported by other studies: mineralized matrix of both alloxan-induced 8-week hyperglycemic rabbit distal femur cortices[44] and STZ-induced 3-week hyperglycemic Swiss TO mouse mid-shaft femur cortices[45] demonstrated a decrease in both hardness and elastic modulus tested by nanoindentation. Reduction of the mechanical properties of hyperglycemic bone has been mainly attributed to the accumulation of AGEs, which occur in the presence of high glucose, change the crosslinking patterns of bone collagen in vivo[46], and decrease of both collagen and OCN production by rat calvarial primary osteoblasts in vitro[47].

In summary, to our knowledge, we have demonstrated, for the first time, that hyperglycemia reduces osteocytic territorial matrix volume due to a reduction in total mineralized matrix volume that results in increased lacunar density in STZ group. Moreover, hyperglycemia decreased the vascular canal volume due to a decrease in canal diameter. However, mean lacunar volume, lacunar sphericity, and vascular canal density and length were not affected. Importantly, not only was the total matrix volume decreased, but the matrix quality was also compromised in STZ compared with the euglycemic group.

## Methods

**Animal Model.** All experimental protocols were approved by the Ethics Committee of Animal Research at the University of Toronto (Protocol # 20011729). Forty-eight male Wistar Albino rats (11-week-old, 200–250 g, Charles River, Canada) were divided into two metabolic groups: Euglycemic and STZ (24 animals per metabolic group). Hyperglycemia was induced via intraperitoneal injection of 65 mg kg$^{-1}$ STZ in the STZ group. Euglycemic animals received an equivalent injection of saline. Glucose levels and body weight were monitored after 24- and 48-hrs post induction. Animals with blood glucose more than 15 mmol L$^{-1}$ in the first 48 h (and maintained thereafter) were considered hyperglycemic. The blood glucose of hyperglycemic rats was measured twice a week using a glucose test strip (FreeStyle Lite, Abbott Diabetes Care, Inc., Alameda, CA, USA), from a drop of blood harvested by tail vein puncture, which was then loaded into a conventional glucometer (FreeStyle Lite). If the level was below 15 mmol L$^{-1}$, STZ was re-injected (maximum twice) intraperitoneally to maintain the hyperglycemic state. All animals were allowed free access to water and rat chow.

**Euthanasia, Sample collection, and Fixation.** Animals were killed either 1 or 3 months after induction of hyperglycemia by cervical dislocation following $CO_2$ exposure. Femora were extracted, cleaned, and kept in either 4% neutral buffered formalin at room temperature for histology and synchrotron micro-CT or in Karnovsky fixative at 4 °C for SEM.

**Femur Length Measurement.** The length (from femoral head to distal condyle) of rat femora from euglycemic and STZ groups was measured using a digital caliper (Mitutoyo, Corp. Kawasaki, Japan). The average length of right and left femora of each animal was used in the analysis.

**Sample Preparation.** Femoral distal metaphyses were investigated in the present study since it is known that the distal growth plate provides the predominant growth in length of the femur, as we have shown previously[48]. Thus, following induction of hyperglycemia in these growing animals, the bone of the distal metaphysis would be hyperglycemic bone compared with euglycemic bone in the control (non-STZ) cohort. To prepare each femoral osteotomy, and measuring from the anterior aspect of the femur, a point was marked 6 mm cranial to the midpoint of the patellofemoral groove. A second point was then marked 4 mm caudal to the first mark. Transverse sections were then cut at each mark to create a plano-parallel 4 mm osteotomy. One batch (n = 6/group) of osteotomies were assigned for histology and SEM (3 bone samples/group/technique) and treated as described below. A second batch (n = 6/group) were further trimmed, to facilitate synchrotron micro-CT examination. First, each of these osteotomies was reduced in height to 2 mm by grinding the cranial surface with silicon carbide papers (P800/

P1200, Buehler, Lake Bluff, IL, USA). Then the medial–lateral periosteal diameter (H), anterior–posterior periosteal diameter (B), medial–lateral endosteal diameter (h), and anterior–posterior endosteal diameter (b) were measured using a digital caliper (Mitutoyo, Corp. Kawasaki, Japan) to allow calculation of the cross-sectional area (A) and MRWT of cortical bone using the formulae below as described previously[49]:

$$A = 3.14(HB - hb)/4 \, (\text{mm}^2)$$

$$MRWT = [(B - b)/b + (H - h)/h]/2$$

Following the measurements, a 2 mm-wide midline longitudinal sample was cut from the anterior aspect of the osteotomy using a cylindrical diamond bur (Brasseler, Québec, QC, Canada) connected to a high-speed system (DCI International, Newberg, OR, USA) (Fig. 8a). These cortical bone pieces were then stored into 4% neutral buffered formalin prior to Micro-CT.

**Histology.** Bone samples were decalcified in Immunocal (Stat Lab, McKinney, TX, USA) for 14 days, dehydrated in a graded ethanol series, and embedded vertically in paraffin blocks. Six-micrometer-thick cross-sections were cut with a Leica microtome (RM2255, Wetzlar, Germany). Cross-sections were then mounted on glass microscope slides, stained with haematoxylin and eosin (hematoxylin and eosin), and imaged with a Leica digital camera (DMC4500, Wetzlar, Germany) attached to a Leica microscope (DMI 4000B, Wetzlar, Germany).

**Scanning Electron Microscopy.** Bone slices were rinsed with cacodylate buffer, dehydrated in a graded ethanol series, and critical point dried in a Bal-tec critical point dryer (CPD030, Columbia Nano Initiative, New York, NY, USA). Samples were then mounted on aluminum stubs, gold coated in a Denton Desk II sputter coater (Leica Microsystems, Wetzlar, Germany), and examined in a Quanta Feg 250 SEM (ThermoFisher Scientific, Hillsboro, OR, USA).

**Synchrotron Radiation Micro-CT.** Imaging: Scanning of the samples was performed using the Biomedical Imaging and Therapy (BMIT) beamline at the Canadian Light Source (Saskatoon, SK, Canada) with a custom Skyscan micro-CT system (Bruker, Kontich, Belgium) at 0.9 μm isotropic voxel size. Projection images were collected with a Hamamatsu CCD camera (C9300–124, Hamamatsu Photonics, Hamamatsu, Japan) fitted with a beam monitor with a 20 μm-thick scintillator.

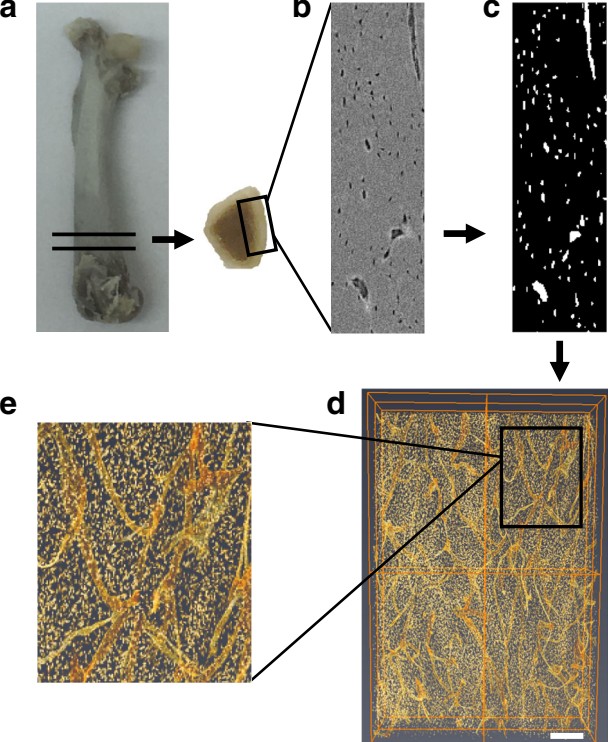

**Fig. 8 Sample preparation and image post-processing. a** Representative image of bone slices cut from the distal femora, **b** original gray-scale image obtained from SR micro-CT, **c** corresponding binarized image, **d, e** 3D volume rendering of the osteocyte lacunae and vascular canals obtained with Volren feature of Amira (scale bar represents 200 μm).

A 40 keV monochromatic beam was employed and the samples were rotated through 180° at a step of 0.2°. Four-frame summing (0.45 s per frame) was used to improve the signal-to-noise ratio. The scan time for each sample was ~1.5 h.

**Image Post-Processing**. The projection images were imported into Nrecon (Bruker, Kontich, Belgium) and reconstructed as approximately 2000 slices (0.9 μm × 2000 slices = 1.8 mm). Reconstructed images were then imported into CT Analyzer 1.16.4.1 (Bruker, Kontich, Belgium), and polygonal volumes of interest (VOIs) were defined, cropped, and saved as new datasets. The new datasets were then imported into Amira 6.3 (ThermoFisher Scientific, Hillsboro, OR, USA) for 3D visualization and quantification of both osteocyte lacunae and vascular canals (Fig. 8b, c). A 3D gaussian filter (2 × 2 × 2) was applied to the images for smoothing and denoising. Volren feature of Amira was used for 3D volume rendering (Fig. 8d, e).

**Osteocyte Lacuna Analysis**. In the stack of gray-scale images, osteocyte lacunae were selected with the interactive thresholding tool based on the difference between gray-scale intensity of osteocyte lacunae (lower intensity, darker) and mineralized bone matrix (higher intensity, brighter). The label analysis tool was then applied to calculate lacuna number, individual lacuna volume ($\mu m^3$), and lacunar sphericity three-dimensionally.

**Vascular Canal Analysis**. Similar to osteocyte lacunae, vascular canals were selected in the stack of gray-scale images with the magic wand tool based on the difference between gray-scale intensity. Thereafter, 3D structure of the vascular canals was visualized in the segmentation editor, and total vascular canal volume ($\mu m^3$) was calculated using the material statistics tool. Visualized vascular canals were then auto-skeletonized in the filament editor to identify vascular canal branching points and to calculate vascular canal segment number, diameter, and length. Vascular density was calculated as vascular canal segment number/VOI. Diameters of vascular canals were derived from the radius values calculated using Amira 6.3. The term segment was used herein to identify a vascular canal length between two branch points or one branch and one end point. Isosurface feature of Amira was used to obtain isometric projections of 3D structure of the vascular canals.

**Mineralized Matrix Volume**. Mineralized matrix volume ($\mu m^3$) was calculated by subtracting the lacunar and vascular canal volume from the VOI. Mineralized matrix volume per osteocyte lacuna ($\mu m^3$) (Osteocytic territorial matrix volume) was calculated by dividing mineralized matrix volume by osteocyte lacuna number. Osteocytic territorial matrix volume is the volume of bone matrix surrounding each lacuna, which has been reported in the previous studies[50,51]. This parameter is particularly important to better understand the mechanobiology of bone, as it is commonly agreed that osteocytes are mechanosensors of bone tissue, and changes in the osteocytic territorial matrix volume affects bone formation and resorption[52]. It is noteworthy that we measured only the mineralized portion of the bone matrix with SR Micro-CT and we calculated the number of osteocyte lacunae, not the osteocyte number, and thus we reference mineralized matrix volume per osteocyte lacuna herein.

**Mechanical Testing**. Nanoindentation was used to evaluate the mechanical properties of femoral cortical bone matrix in euglycemic and hyperglycemic rat femora. Bone samples used for SR Micro-CT were embedded in acrylic resin (GC America, Inc., Alsip, IL, USA), ground, and polished to a 0.25 μm finish with diamond paste (Hudson supply, Cleveland, OH, USA). The bone surface was visualized with an area scan camera (UI-155xLE, IDS, Obersulm, Germany) attached to a microscope built into a PB1000 Nanovea nanoindentation system (Nanovea Inc., Irvine, CA, USA). Eight indentations, away from vascular canals, osteocyte lacunae, and microcracks, were then performed using the system equipped with a Berkowitch diamond probe. Indentations were made up to a depth of 946 nm with a loading/unloading rate of 24 mN min$^{-1}$ and an approach speed of 5 μm min$^{-1}$. At maximum load (10 mN), a holding period of 20 s was applied to avoid creeping of the bone material. Hardness and elastic modulus of mineralized matrix were determined by Nanovea nano hardness tester software (Nanovea Inc., Irvine, CA, USA).

**Statistics and Reproducibility**. Unless stated otherwise, dependant variables were analyzed using GraphPad Prism 6 (GraphPad Software, Inc., San Diego, CA, USA). Means for each dependent variable were compared using a two-way analysis of variance using time and animal condition as fixed effects and the interaction term included. When significant, the Newman–Keuls test was utilized for post-hoc analysis.

In the case of osteocyte lacunar density on histology sections, Student's $t$-test was used for analysis of the results as there was only data available from two groups. In addition, the dependent variables total canal volume/VOI, mean segment volume, and mean segment diameter were found to have significantly differing variances as determined by the Bartlett test for homogeneity of variance. These dependent variables were then analyzed using R statistical software (version 3.4.0). The variables were transformed based on information obtained from the Box-Cox transformation. Total canal volume/VOI and mean canal segment volume were transformed by raising each outcome to the power of −0.5 and the mean canal segment diameter was transformed by raising each outcome to the power of −2. Following this homogeneity of variance for each group was rechecked using the Bartlett test and significant difference between the groups was determined by linear modeling, using time and animal condition as fixed effects and including the interaction term.

All residuals were confirmed not to differ significantly from a normal distribution using the Kolmogorov-Smirnov test. Significance is indicated as follows: *$p < 0.05$, **$p < 0.01$, ***$p < 0.001$, and ****$p < 0.0001$. The results are expressed as the mean ± SD.

**Reporting summary**. Further information on research design is available in the Nature Research Reporting Summary linked to this article.

## Data availability
The majority of the raw data were generated at the Canadian Light Source large-scale facility. Source data for figures are provided with the paper as Supplementary Data.

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

## Acknowledgements

This work was partially funded by ZimmerBiomet Dental for which the authors acknowledge support [grant number: 493601]. A part of the research described in this paper was performed at the Canadian Light Source (CLS), which is supported by the Canada Foundation for Innovation, Natural Sciences and Engineering Research Council of Canada, the University of Saskatchewan, the Government of Saskatchewan, Western Economic Diversification Canada, the National Research Council Canada, and the Canadian Institutes of Health Research. The assistance of Denise Miller with scanning operations (at CLS) and Madelaine Jong with data analysis (at UofT) is greatly appreciated. B.A. acknowledges the Ministry of National Education, Republic of Turkey for his post-graduate scholarship.

## Author Contributions

B.A. and J.E.D. designed the experiments. B.A., R.S.L., and J.E.D. performed the synchrotron micro-CT scanning and image post-processing. B.A., K.P., and D.M.L.C. analyzed the synchrotron micro-CT data. B.A. conducted the SEM and histology sample preparation and imaging. B.A., Y.Q., and G.G. planned and conducted the mechanical tests. B.A. and J.E.D. wrote the manuscript. B.A., J.E.D., and D.M.L.C. made manuscript revisions. B.A. takes responsibility for the integrity of the data analysis. All authors have approved the final article.

## Competing interests

The authors declare no competing interest.

## Additional information

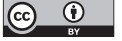

