## [Peer Review File · Communications Biology]

Reviewers' comments:

Reviewer #1 (Remarks to the Author):

General comment:

In this manuscript, the authors investigated the effect of hyperglycemia on bone matrix quality by calculating various measurements including the osteocyte's territorial matrix volume, lacunar density, lacunar volume, lacunar sphericity, vascular canal density and length, and mechanical properties. The topic of the manuscript is interesting to the experts who are related in the fields and the significance of this study seems high but there are several points that should be clarified and modified before publication. In addition, this paper must be carefully self-reviewed by the authors and re-written because the manuscript has a wrong figure number, probably during the final editing process. Some images need to be improved in terms of resolution and better readability.

Some main points need to be clarified and modified are:

1. Some figures are inappropriately numbered: Figure 5 and 6 seemed to be merged, therefore, there is no Figure 8 in the submitted manuscript. This needs to be carefully reviewed and fixed by the authors and caused a major revision.
2. The sample preparation process and image analysis method for the SEM seemed to be inappropriate. For the SEM image analysis shown in Figure 3a, the samples should be embedded in a resin and then polished appropriately in order to represent a flat surface. In general, the freeze-fractured surface showed the high surface roughness, resulting in different height of sample locations in an image, thus, the number of lacunar measurement of this sample is less accurate than the one that has flat surface due to the topological effect. This needs to be clarified or justified.
3. For Figure 4, the authors seemed to use "isosurface" feature in the Amira Software, which is based on the selected, specific intensity. Can the authors also add "3D volume rendering" images to show a better comparison among the images in order to avoid any subjective selection? The rendering images obtained by "Isosurface" feature without the histogram profile or other volume rendering images cannot be a solid representation by itself.
4. Discussions: There are too many discussions which are not directly related to the present work. Please carefully review and remove unnecessary sentences which are speculating the author's opinion without direct evidences from the results of this work.

Reviewer #2 (Remarks to the Author):

This is a well written and well done study focusing on altered bone quality that contributes to diabetic skeletal fragility. More specifically, the authors measured osteocyte lacunar volume and vascular canal volume in STZ-treated rodents to elucidate cellular and vascular mechanisms due to hyperglycemia. I have a few minor comments:

Authors use an STZ model, which is not necessarily the best animal model to use (see paper by Fajardo, JBMR, 2014). Can the authors comment on the reasons for using this particular model?

Line 54 - authors refer to hyperglycemic subjects but cites an animal study, there are 2-3 recent studies that measure AGEs in humans that should be referenced

Discussion section - I suggest the authors start off with an "intro" paragraph to reiterate the motivation/goal of the study and the major finding, before diving into comparisons with previous studies

Reviewer #3 (Remarks to the Author):

The Authors compared bone quality between euglycemic and streptozotocin-induced hyperglycemic (STZ) from different aspects: (1) cortical bone dimensions of specimen, (2) lacunae and vascular analysis from histology and high resolution synchrotron micro-CT, and (3) mechanical testing on specimen. Their results demonstrated that the rats in STZ group has (1) less cortical thickness and area, (2) more lacunars, (3) smaller vascular canal, and (4) poorer cortical bone quality. The study was completely well done, the experiment were fitted their study purpose, and the steps of analysis were logical. I recommend to published their work after they revised the following few points in this manuscript,

Points:

(A) Result

(a) (Page 6, Line 123-124; Fig. 5c) It is not very clear what "Vascular canal segment number/VOI" means. Does it mean "vascular density"?

(b) (Page 7, Line 137 -142) In the discussion section, you mentioned the relationship between vessel volume and bone forming. It will be an useful evidence for this if you could analyze correlation between your vascular parameters (eg. vascular canal volume) and the mechanical parameters.

(B) Discussion

(a)(Page 11, Line 242-245) Do you have any markers to show the dynamic change of bone in you study?

(C) Materials and Methods:

(a)(Page 13, Line 276-277) Why you compare data between "1 month" and "3 months" but not any other period? If there is any specific reason for choosing these periods, please describe it in the method or discussion section.

(b)(Page 15, Line 320) Is it isotropic? If yes, please clarify in the statement.

(c)(Page 16, Line 335; Line 340) I suggest the Authors add example images about their thresholding method for lacunae and vascular canal segmentation. For example, you could show the original gray-scale image and histological image at the same part of bone at first, and then show the binary mask after segmentation. This will be helpful to demonstrate that your method was actually successful to detect lacunae and vascular canal.

Response to the Reviewers' comments:

Reviewer #1 (Remarks to the Author):

General comment:

In this manuscript, the authors investigated the effect of hyperglycemia on bone matrix quality by calculating various measurements including the osteocyte's territorial matrix volume, lacunar density, lacunar volume, lacunar sphericity, vascular canal density and length, and mechanical properties. The topic of the manuscript is interesting to the experts who are related in the fields and the significance of this study seems high but there are several points that should be clarified and modified before publication. In addition, this paper must be carefully self-reviewed by the authors and re-written because the manuscript has a wrong figure number, probably during the final editing process. Some images need to be improved in terms of resolution and better readability.

Some main points need to be clarified and modified are:

1. Some figures are inappropriately numbered: Figure 5 and 6 seemed to be merged, therefore, there is no Figure 8 in the submitted manuscript. This needs to be carefully reviewed and fixed by the authors and caused a major revision.

Answer: Figure 5 and 6 were separately uploaded to the manuscript tracking system and separately mentioned in the "Results" section of the manuscript. Therefore, we believe that the reason why Figure 5 and Figure 6 look merged is related to the way that the manuscript tracking system creates the final pdf. We also reviewed the numbers of all figures, and they were indeed appropriately numbered.

2. The sample preparation process and image analysis method for the SEM seemed to be inappropriate. For the SEM image analysis shown in Figure 3a, the samples should be embedded in a resin and then polished appropriately in order to represent a flat surface. In general, the freeze-fractured surface showed the high surface roughness, resulting in different height of sample locations in an image, thus, the number of lacunar measurement of this sample is less accurate than the one that has flat surface due to the topological effect. This needs to be clarified or justified.

Answer: The reviewer raises an important point. We employed freeze-fractured samples to enable comparison with our previously published data (text Ref #11). However, to address the reviewer's legitimate concern, we have now also calculated osteocyte lacuna number/bone area in the histological sections of demineralized bone, and we have compared these values with those obtained from the freeze-fractured surfaces. Similarly to the osteocyte lacuna number/bone area in the SEM images of freeze fractured surfaces (Fig. 3c – a 19% increase in the mean), osteocyte lacuna number/bone area in histological sections was significantly higher in STZ compared to euglycemic group (Fig. 3d – also an 18.7 increase in the mean). Therefore, we believe that the freeze fracturing process did not significantly impede the visualization of osteocyte lacunae in the present study since the trend of the data and the osteocyte

lacuna number/bone area values were comparable in both freeze-fractured surfaces and demineralized histological sections. Nevertheless, we have made the following addition to the text: [Increased lacunar density was visualized in the SEM images of freeze fractured specimens (Fig. 3a) and histological sections of demineralized rat femora (Fig. 3b) in STZ group. Indeed, osteocyte lacuna number/bone area was statistically significantly higher in STZ group at both 1- and 3-months when the osteocyte lacunae were counted manually and normalized to the bone area in the SEM images at the same magnification (Fig. 3c). **Similarly, osteocyte lacuna number/bone area in the histological sections was statistically significantly greater in STZ compared to euglycemic group (Fig. 3d). Increased osteocyte lacunar density in hyperglycemic bone is illustrated in Fig. 3e.]**

3. For Figure 4, the authors seemed to use “isosurface” feature in the Amira Software, which is based on the selected, specific intensity. Can the authors also add “3D volume rendering” images to show a better comparison among the images in order to avoid any subjective selection? The rendering images obtained by “Isosurface” feature without the histogram profile or other volume rendering images cannot be a solid representation by itself.

Answer: As the reviewer suggested, in the revised version of our manuscript, we have included 3D volume rendering images with the isometric projections of 3D reconstructions for each sample in Figure 4 to avoid any subjective selection.

4. Discussions: There are too many discussions which are not directly related to the present work. Please carefully review and remove unnecessary sentences which are speculating the author’s opinion without direct evidences from the results of this work.

Answer: We have reviewed our discussion and removed the paragraphs below from the Discussion:

[Even though loss of insulin could be considered as a contributing factor to decreased matrix production by osteoblasts in an STZ-induced hyperglycemic rat model, it has been reported that hyperglycemia decreases matrix formation in osteogenic cultures containing fetal bovine serum (FBS)^{28,29}, which provides low amounts of insulin. Thus, we assume that the decrease in mineralized matrix volume demonstrated in the present study was mainly affected by hyperglycemia rather than insulin loss.]

[It is generally agreed that osteocytes are mechanosensors of bone, and produce the signaling molecules such as nitric oxide and prostaglandins, which can regulate the activity of both osteoclasts and osteoblasts³⁴. In the current study we have demonstrated, for the first time, that osteocytic territorial matrix significantly decreased in the femoral cortices of STZ-induced hyperglycemic rats (Fig. 6b and illustrated in Fig. 3e). Therefore, the decrease in osteocyte territorial matrix volume in the STZ group needs to be investigated in human subjects since one can speculate that osteocytes with smaller territorial matrix are more likely to sense the changes in their microenvironment and might accelerate bone turnover in hyperglycemic individuals, which would then result in bone loss due to increased osteoclast resorption and decreased matrix production by osteoblasts.]

[Therefore, one can speculate that since there would be more bone resorption and/or less bone formation in hyperglycemic individuals, cortical pores, which are larger than both vascular canals and osteocyte lacunae, would form and increase the total cortical porosity.]

Reviewer #2 (Remarks to the Author):

This is a well written and well done study focusing on altered bone quality that contributes to diabetic skeletal fragility. More specifically, the authors measured osteocyte lacunar volume and vascular canal volume in STZ-treated rodents to elucidate cellular and vascular mechanisms due to hyperglycemia. I have a few minor comments:

We thank the reviewer for these kind remarks.

Authors use an STZ model, which is not necessarily the best animal model to use (see paper by Fajardo, JBMR, 2014). Can the authors comment on the reasons for using this particular model?

Answer: We had two reasons to use STZ-induced rat model in the present study: First, we initially observed the increased cellularity in hyperglycemic bone in STZ-induced hyperglycemic rat model (text Ref #23). Therefore, since we would like to further investigate increased cellularity in hyperglycemic bone, we used the same animal model in the present study. Second, as we mentioned in both our Introduction and Discussion, in the studies by Kerckhofs et al. (2016) (text Ref #20) and Karunaratne et al. (2016) (text Ref #21), the authors investigated the changes in lacunar and vascular canal parameters in the animal models with multiple pathologies. For instance, Karunaratne et al. (2016) employed hyperglycemic $Crh^{-120/+}$ mice which were also osteoporotic and exhibited considerable interconnected porosity, which clearly contributed to decreased lacunar density reported in their paper. Therefore, from their results, it is difficult to attribute the changes observed in bony structure to either hyperglycemia or osteoporosis independently. In this context, an advantage of using STZ-induced hyperglycemic rat model is to avoid having multiple pathologies, such as obesity and osteoporosis, which might shadow the effects of hyperglycemia on bony structure. Taken together, as stated in the Fajardo et al. (2014) (text Ref #22), we agree that “It is likely that no single animal model will recapitulate all of the features of diabetic skeletal fragility in humans”, and we have added this comment to our Introduction: [Since $Crh^{-120/+}$ mice are obese, hypercorticosteronaemic, and hyperglycemic, it is difficult to attribute the changes observed in bony structure to either hyperglycemia or osteoporosis independently. **Even though there is no single animal model that can show all the features of diabetic skeletal fragility in humans as previously discussed by Fajardo et al.²²,** the effects of metabolic changes on bone microstructure need to be investigated in a simpler animal model].

Line 54 - authors refer to hyperglycemic subjects but cites an animal study, there are 2-3 recent studies that measure AGEs in humans that should be referenced.

Answer: We thank the reviewer for pointing this out, and we agree that adding some human clinical references would be advantageous. Thus, we have added the following 2 references that address such human data:
We have added references to our Introduction:"

[It is generally agreed that bone quality is compromised in hyperglycemic compared to euglycemic bone⁴⁻⁷, but the reasons are poorly understood. Reports of decreased implant stability⁸ and retention^{9,10} in hyperglycemic subjects support the notion of compromised bone quality. It has been demonstrated that bone healing delays¹¹, growth plate thickness reduces¹², cortical porosity increases due to bone loss^{13,14}, and the crosslinking patterns of bone collagen changes with advanced glycation end products (AGEs)^{15,16,17} in hyperglycemic subjects. Yet, little has been done to elucidate the changes in bone cell density and vascular architecture in hyperglycemic bone.]

Text ref #16. Saito, M., Fujii, K., Soshi, S. & Tanaka, T. Reductions in degree of mineralization and enzymatic collagen cross-links and increases in glycation-induced pentosidine in the femoral neck cortex in cases of femoral neck fracture. *Osteoporos. Int.* 17, 986-995 (2006).

Text ref #17. Karim, L. et al. Bone microarchitecture, biomechanical properties, and advanced glycation end-products in the proximal femur of adults with type 2 diabetes. *Bone* 114, 32-39 (2018).

Discussion section - I suggest the authors start off with an "intro" paragraph to reiterate the motivation/goal of the study and the major finding, before diving into comparisons with previous studies

Answer: An introduction paragraph has been added into the "Discussion".

Reviewer #3 (Remarks to the Author):

The Authors compared bone quality between euglycemic and streptozotocin-induced hyperglycemic (STZ) from different aspects: (1) cortical bone dimensions of specimen, (2) lacunae and vascular analysis from histology and high resolution synchrotron micro-CT, and (3) mechanical testing on specimen. Their results demonstrated that the rats in STZ group has (1) less cortical thickness and area, (2) more lacunars, (3) smaller vascular canal, and (4) poorer cortical bone quality. The study was completely well done, the experiment were fitted their study purpose, and the steps of analysis were logical. I recommend to published their work after they revised the following few points in this manuscript,

Points:

(A) Result

(a) (Page 6, Line 123-124; Fig. 5c) It is not very clear what “Vascular canal segment number/VOI” means. Does it mean “vascular density”?

Answer: Yes, “vascular canal segment number/VOI” means “vascular density”. Thus, we have changed it to “vascular density” in the revised manuscript to make this parameter clearer to the readers. In addition, a sentence which explains how we calculated the vascular density has been added to the “Vascular Canal Analysis” section in Materials and Methods.

(b) (Page 7, Line 137 -142) In the discussion section, you mentioned the relationship between vessel volume and bone forming. It will be an useful evidence for this if you could analyze correlation between your vascular parameters (eg. vascular canal volume) and the mechanical parameters.

Answer: In our Discussion, we mentioned that decreased vascular canal volume could be due to decreased blood vessel volume, which might result in decreased bone formation by osteoblasts. This does not mean that the matrix mechanical parameters of hyperglycemic bone are lower in our study because of the decreased bone formation. Indeed, it is known that regardless of the quantity, bone quality could be compromised in hyperglycemic subjects (Yamaguchi and Sugimoto 2012). In this context, we did not claim the existence of a relation between lower vascular canal volume and decreased matrix quality. Instead, we discussed that decreased vascular canal volume could be a potential reason for decreased bone formation in hyperglycemic bone.

Reference

Yamaguchi, T. & Sugimoto, T. Bone metabolism and fracture risk in type 2 diabetes mellitus. Bonekey Rep. 7, 1–36 (2012).

(B) Discussion

(a)(Page 11, Line 242-245) Do you have any markers to show the dynamic change of bone in you study?

Answer: We do not have markers to show in the present study, but we have used the STZ-induced hyperglycemic rat model in our laboratory for more than 5 years. Therefore, in our previous paper published in Acta Biomaterialia, sequential fluorochrome labeling with alizarin red and tetracycline hydrochloride demonstrated that significantly less mineralized bone formed in the defect area, created in the distal cortex of rat femora, in hyperglycemic compared to euglycemic group 30 days after surgery (Text ref #11).

(C) Materials and Methods:

(a)(Page 13, Line 276-277) Why you compare data between “1 month” and “3 months” but not

any other period? If there is any specific reason for choosing these periods, please describe it in the method or discussion section.

Answer: We thank the reviewer for raising this point.

In one of our recent studies (yet to be published), we investigated the osteointegration of the titanium implants in euglycemic and hyperglycemic (STZ) rats using a mechanical disruption test. In the aforementioned study, osseointegration was achieved by 1 month, with small differences between groups. But the difference in disruption force between groups was more pronounced after 3 months (a time period that included peri-implant bone remodelling). Therefore, in the present study, we chose 1 month and 3 months to investigate the changes in bone microstructure in these unique time points in hyperglycemic state.

Thus, we have now added the abbreviated explanation below into our Introduction:-

Based on our previous studies of implant osseointegration, we chose 2 time points: 1 month, by which time bone healing is complete,²³ and 3 months by which time significant bone remodeling will have occurred (unpublished data).

(b)(Page 15, Line 320) Is it isotropic? If yes, please clarify in the statement.

Answer: Yes, our voxel size is isotropic. The term “isotropic” is now added into the related sentence in the “Synchrotron Radiation (SR) Micro-CT” section in Materials and Methods.

(c)(Page 16, Line 335; Line 340) I suggest the Authors add example images about their thresholding method for lacunae and vascular canal segmentation. For example, you could show the original gray-scale image and histological image at the same part of bone at first, and then show the binary mask after segmentation. This will be helpful to demonstrate that your method was actually successful to detect lacunae and vascular canal.

Answer: Figure 8 has been revised based on the reviewer’s suggestion. We added gray-scale micro-CT and corresponding binarized images in Figure 8 along with 3D volume rendering images of the same sample.

REVIEWERS' COMMENTS:

Reviewer #1 (Remarks to the Author):

The revised manuscript looks good enough to be published as it is.

Reviewer #2 (Remarks to the Author):

The authors have adequately addressed my prior comments/concerns.

Reviewer #3 (Remarks to the Author):

After the revision, the Authors has clarified the confused points in their article. I recommend to publish this article.